# Serum Soluble Lectin-like Oxidized Low-Density Lipoprotein Receptor-1 (sLOX-1) Is Associated with Atherosclerosis Severity in Coronary Artery Disease

**DOI:** 10.3390/biom13081187

**Published:** 2023-07-29

**Authors:** Katharine A. Kott, Elijah Genetzakis, Michael P. Gray, Peter Hansen, Helen M. McGuire, Jean Y. Yang, Stuart M. Grieve, Stephen T. Vernon, Gemma A. Figtree

**Affiliations:** 1Cardiovascular Discovery Group, Kolling Institute of Medical Research, University of Sydney, St Leonards, NSW 2065, Australia; katharine.kott@sydney.edu.au (K.A.K.);; 2Department of Cardiology, Royal North Shore Hospital, St Leonards, NSW 2065, Australia; 3School of Medical Sciences, Faculty of Medicine and Health, University of Sydney, Camperdown, NSW 2006, Australia; 4Ramaciotti Facility for Human Systems Biology, University of Sydney, Camperdown, NSW 2006, Australia; 5School of Mathematics and Statistics, University of Sydney, Camperdown, NSW 2006, Australia; 6Imaging and Phenotyping Laboratory, Charles Perkins Centre, Faculty of Medicine and Health, University of Sydney, Camperdown, NSW 2006, Australia; 7Department of Radiology, Royal Prince Alfred Hospital, Camperdown, NSW 2006, Australia

**Keywords:** atherosclerosis, computed tomography coronary angiography (CTCA), coronary artery disease (CAD), biomarker, soluble lectin-like oxidized low-density lipoprotein receptor-1 (sLOX-1)

## Abstract

Risk-factor-based scoring systems for atherosclerotic coronary artery disease (CAD) remain concerningly inaccurate at the level of the individual and would benefit from the addition of biomarkers that correlate with atherosclerosis burden directly. We hypothesized that serum soluble lectin-like oxidized low-density lipoprotein receptor-1 (sLOX-1) would be independently associated with CAD and investigated this in the BioHEART study using 968 participants with CT coronary angiograms, which were scored for disease burden in the form of coronary artery calcium scores (CACS), Gensini scores, and a semi-quantitative soft-plaque score (SPS). Serum sLOX-1 was assessed by ELISA and was incorporated into regression models for disease severity and incidence. We demonstrate that sLOX-1 is associated with an improvement in the prediction of CAD severity when scored by Gensini or SPS, but not CACS. sLOX-1 also significantly improved the prediction of the incidence of obstructive CAD, defined as stenosis in any vessel >75%. The predictive value of sLOX-1 was significantly greater in the subgroup of patients who did not have any of the standard modifiable cardiovascular risk factors (SMuRFs). sLOX-1 is associated with CAD severity and is the first biomarker shown to have utility for risk prediction in the SMuRFless population.

## 1. Introduction

Cardiovascular disease secondary to atherosclerosis remains the highest cause of mortality in both sexes worldwide [1], and while the identification and treatment of standard modifiable cardiovascular risk factors (SMuRFs) and the implementation of optimal medical therapy has significantly improved outcomes [2,3,4], current risk scoring algorithms for coronary artery disease (CAD) prediction have high margins of error in both directions [5]. Blood-based biomarkers specific to atherosclerosis have the potential to substantially improve risk stratification when added to traditional risk scoring systems and may help identify the significant proportion of patients demonstrated in recent national and international studies [6,7,8,9,10,11] who have been found to present with the acute coronary syndrome (ACS) despite having none of the SMuRFs.

A candidate biomarker for this predictive role is the lectin-like oxidized low-density lipoprotein receptor-1 (LOX-1), which was first described in 1997 as a scavenger receptor for modified low-density lipoprotein (LDL) expressed by vascular endothelium [12]. This receptor has been extensively studied in subsequent years and is now understood to be a critical player in the pathophysiology of atherosclerosis, with roles as a mediator of angiogenesis, platelet activation, leukocyte adhesion, endothelial dysfunction, fibroblast, and smooth muscle cell proliferation, and in foam cell formation [13,14,15]. The soluble form of LOX-1 (sLOX-1) is readily measured in serum samples and is derived from cleavage of LOX-1 from the cell surface in response to pro-inflammatory stimuli such as oxidized-LDL, C-reactive protein, tumor necrosis factor α, and interleukins 8 and 18 [16,17,18].

LOX-1 is expressed at high levels on the surface of unstable plaques in both animal models [19] and in humans [20], and during invasive angiography serum, sLOX-1 levels in the coronary sinus were significantly higher than in matched samples from the aorta, strongly suggesting that sLOX-1 originates from the plaque and is released into the coronary circulation [21]. Indeed, elevated levels of circulating sLOX-1 have been demonstrated to be an early marker of ACS [22,23,24], and optical coherence tomography testing confirms even higher levels of sLOX-1 in the context of ACS with plaque rupture [25]. In studies of patients undergoing angiography for ACS or symptomatic angina, sLOX-1 levels were found to be higher in patients with more proximal disease [26] and with more complex lesions [27], suggesting that sLOX-1 may correlate with disease burden and instability. In patients with quiescent moderate CAD diagnosed on previous angiography, sLOX-1 was shown to be an independent predictor of major adverse cardiac events (MACE) at two years and was not independently associated with any of the major cardiovascular risk factors or clinical variables [28].

Based on the known biology of LOX-1 and the results of previous clinical studies, we hypothesized that sLOX-1 may be a biomarker with prognostic value for the burden of atherosclerotic plaque present within a given individual. To investigate this relationship, we included 968 participants from the BioHEART study [29] who have undergone clinically indicated computed tomography coronary angiography (CTCA) for suspected CAD, allowing for detailed scoring of disease burden and assessment of plaque morphology. Of the 636 patients who had CAD identified on CTCA, 15% were also SMuRFless. CTCA additionally offers the advantage of clearly distinguishing the group of patients who did not have detectable CAD, providing an accurately defined negative control group. As a positive assay control, we included 106 patients from another BioHEART cohort who had angiographically confirmed ACS.

## 2. Materials and Methods

### 2.1. Study Design

The protocol for the BioHEART biobank (Australia New Zealand Clinical Trials Registry ANZTR12618001322224) has been described in detail previously [29]. The study design for this biomarker analysis is shown in Figure 1 and is also described briefly in the following sections.

### 2.2. Participants and Clinical Data

BioHEART-CT is a sub-study of the BioHEART study. Adult patients who were referred for a clinically indicated CTCA for suspected CAD were recruited for inclusion in the BioHEART study from 2015 onwards. Patients were included if they were able to provide informed consent and were age 18 or older. Patients who were unwilling or unable to participate in follow-up were excluded. All participants were recruited as outpatients from an imaging facility associated with a large tertiary hospital in Sydney, Australia.

Baseline clinical data were collected at the time of recruitment using a data collection form. Data collected included self-reported demographic data, smoking history, alcohol intake, past medical history, medication list, history of recent cardiac symptoms, and the indication for the CTCA. Current smoking was defined as having smoked tobacco products regularly in the last 12 months, and a significant smoking history was defined as 10 or more pack years. The major standard modifiable cardiovascular risk factors (SMuRFs) were defined as self-reported diagnosis or medical therapy for hypertension, hyperlipidemia, diabetes mellitus, and significant smoking history. A significant family history of ischemic heart disease was recorded if it was reported in a first-degree relative who was under 60 years of age.

The BioHEART-CT discovery cohort includes the first 1000 patients recruited to the study who had technically adequate CTCAs, sufficient stored blood samples for planned biomarker discovery work, and who did not have a history of coronary artery graft surgery or previous cardiac stenting. Of these, 968 were included in this analysis, as 32 had insufficient stored serum samples or were omitted because of technical issues during the assay.

The ACS cohort was taken from BioHEART-MI, a sub-study of BioHEART. Blood samples were collected at the time of index angiography. Patients were included if they were age 18 or greater and were willing and able to provide informed consent. Patients were excluded if they were highly dependent on medical care, were unwilling or unable to participate in ongoing follow-up, or had a non-ischemic cause for their presentation. Baseline clinical information was obtained from a review of the medical records and angiograms. Patients used in this study had an ECG demonstrating ischemic changes with a culprit lesion identified on angiography. A total of 106 patients with ACS were included in this analysis.

### 2.3. CT Imaging Analysis

CTCA scans were performed on a Phillips iCT 256 slice scanner, and Phillips iPatient software was used for the creation of reconstructions. All scans were protocolled by a radiologist with a minimum of level two accreditation, and all radiographers had detailed training for CTCA workup and acquisition. Heart rate optimization was achieved by giving oral metoprolol, or ivabradine if beta-blockers were contraindicated. If the heart rate was sufficiently controlled, the study was performed prospectively, otherwise retrospective acquisition was utilized. Oral nitroglycerine (600–800 micrograms) was given immediately prior to intravenous contrast injection. Radiation dose was minimized in line with current recommendations [30].

Imaging data were exported as thin DICOMs, which were de-identified and stored securely for later analysis. Coronary artery calcium scores (CACS) were generated using the Agatston method [31] via the Phillips IntelliSpace Portal 8.0. The total calcium score and the demographic-matched calcium percentile scores (Ca%) [32] were recorded. CTCAs were analyzed by the 17-segment standard model recommended by the Society of Cardiovascular Computed Tomography [33]. Analysis of the segments and scoring of stenosis was performed as per Figure 2, based on the Gensini scoring system [34]. An additional assessment of the composition of the plaque in each segment (calcified, soft, or mixed) was performed allowing further calculation of a modified Gensini score. A soft-plaque score (SPS) was generated by subtracting the modified Gensini score from the Gensini score. A soft-plaque score ratio (SPSr) was created by dividing the soft-plaque score by the Gensini score. To phenotype the cohort, patients were categorized into groups as follows: no detectable CAD (CAD-): Gensini = 0, Calcified Plaque Predominant: Gensini > 0, SPSr < 1, Soft Plaque Predominant: Gensini > 0, SPSr ≤ 1.

### 2.4. Biological Samples and Analysis

Blood samples used for sLOX-1 measurement were collected into serum pathology tubes (BD, North Ryde, Australia) and stored on ice until processing. Tubes were centrifuged at 1861× *g* for 15 min at 4 °C. Aliquots of serum were frozen and stored at −80 °C until analysis. Frozen samples were thawed at room temperature, and serum levels of sLOX-1 were assessed by ELISA assay utilizing a Human LOX-1 ELISA kit (Abcam, Melbourne, Australia—kit #ab212161) according to manufacturer instructions.

### 2.5. Statistical Analysis

The BioHEART-CT patients were divided into subgroups based on imaging scores as described above. Categorical variables are presented using frequencies and percentages, and numerical variables are described using means, standard deviations, medians, and interquartile ranges as appropriate. Comparison between the subgroups was performed using unpaired *t* tests, with a 2-tailed *p* value < 0.05 considered statistically significant. Bivariate correlations of continuous variables are presented as Pearson Correlation Coefficients with associated 2-tailed *p* values.

Linear regression models were created using the independent variables of age, sex, body mass index (BMI), hypertension, hyperlipidemia, diabetes mellitus, significant smoking history, and significant family history of premature CAD, with and without sLOX-1. The model was repeated using each of the three disease scores as the dependent variable (Gensini, CACS, and SPS), and then re-run using only the patients who had no SMuRFs (“SMuRFless”). The model R^2^, adjusted R^2^, F values, *p* values, and *p* values for model change were calculated for each model, with a comparison made between models with and without sLOX-1.

Logistic regression models were created using the same independent variables and the dependent variables of any detectable CAD (Gensini > 0), moderately obstructive CAD (stenosis > 50% in any coronary artery), and severely obstructive CAD (stenosis > 75% in any coronary artery). Again, the model was repeated using only the SMuRFless patients as a subgroup analysis.

All data were analyzed in IBM SPSS Statistics (version 28.0.0.0) and graphed using GraphPad Prism (version 9.5.1).

## 3. Results

### 3.1. Clinical Characteristics and Disease Burden

A total of 968 patients were included from the CTCA cohort, and, of this number, 333 had no detectable CAD, 309 had the calcified plaque predominant phenotype, and 327 had the soft-plaque predominant phenotype. Another 106 patients were included from the ACS cohort as positive controls. Table 1 outlines the clinical characteristics and disease burden of the various subgroups.

The most common cardiovascular risk factor in the cohort was hyperlipidemia (59.1%), followed by hypertension (39.2%), significant smoking history (19.4%), and diabetes mellitus (8.8%). Approximately one-fifth of the cohort had a significant family history of premature CAD (20.8%), a tenth had atrial fibrillation (12.1%) or inflammatory arthritis (10.7%), and smaller numbers had a previous stroke or transient ischemic attack (TIA, 5.2%), or peripheral arterial disease (1.4%). Cardiac risk factor prevalence was higher in the disease phenotypes. Cardiac medications being taken in the cohort included statins (33.4%), ACE inhibitors or angiotensin receptor blockers (31.9%), anti-platelet medications (17.1%), beta blockers (14.4%), and anticoagulants (9.0%), again with a higher prevalence of these being taken within the disease subgroups. When compared to the calcified-plaque predominant phenotype, more of the patients with the soft-plaque predominant phenotype were taking anti-platelet agents (22.9% vs. 16.8%) and statins (48.3% vs. 34.0%). The subgroup that had no detectable CAD (CAD-) was younger than those with detectable disease (average 53 vs. 65 years), was more frequently female, and had lower rates of all coronary risk factors and medications that were assessed.

Across the CTCA cohort participants had a median Gensini score of 3.5, a median soft-plaque score of 2.5, a median CACS of 9.9, and an average calcium percentile of 38.2%. Moderate obstructive disease, as defined by stenosis > 50% in any vessel, was observed in 18.3% of the total cohort, 20.4% of the calcified-plaque predominant phenotype, and 34.9% of the soft-plaque predominant phenotype. Severe obstructive disease was defined as stenosis > 75% in any vessel, and was present in 5.6% of the total cohort, being found in 4.9% and 11.9% of the calcified and soft-plaque predominant phenotypes, respectively. All disease scores were higher in the soft-plaque predominant subgroup when compared to the calcified-plaque predominant subgroup.

When compared to the CTCA group as a whole, the ACS group had a higher percentage of males (71.4% vs. 55.7%), had more patients with three of the four SMuRFs (19.0% vs. 8.5%), had more diabetics (17.1% vs. 8.8%), and more patients with a significant smoking history (37.1% vs. 19.4%). Disease scores were not calculated for the ACS group, who were all confirmed to have culprit acute plaque rupture—and thus by definition obstructive disease—as part of their inclusion criteria.

### 3.2. sLOX-1 Levels in Stable CAD

Mean sLOX-1 values for each subgroup are shown in Table 1, and the distribution of values for each subgroup is demonstrated in Figure 3. Serum sLOX-1 levels did not significantly differ between the CAD- subgroup and either of the stable disease groups. Consistent with prior studies [22,23,25], sLOX-1 levels in ACS patients were significantly higher than in the stable patients who had not suffered an infarction (*p* < 0.001).

### 3.3. sLOX-1 Is Predictive of Disease Severity

To determine the relationship of sLOX-1 with disease severity, multivariable linear regression modeling was performed with independent variables of age, sex, body mass index (BMI), hypertension, hyperlipidemia, diabetes mellitus, significant smoking history, significant family history, and sLOX-1 level. The model was repeated with each of the disease scores (Gensini, CACS, and SPS) as the dependent variable. The standardized beta coefficients and associated *p* values are shown in Table 2, and the change in model R^2^ is shown in Table 3 and Figure 4 (orange bars). Across the whole cohort, the addition of sLOX-1 to the base model significantly improved the prediction of both the Gensini score and SPS (β = 0.062, *p* = 0.025 and β = 0.072, *p* = 0.015, respectively). Model R^2^ improved from 0.284 to 0.288 for the Gensini score and 0.175 to 0.180 for the SPS score. No relationship was detected for CACS (β = 0.047, *p* = 0.110).

The relationship between sLOX-1 and CAD incidence was assessed in logistic regression models, which incorporated the same independent variables used in the linear regression. As shown in Table 4 and Figure 5, the odds ratio for sLOX-1 was not significant for the prediction of the presence of CAD as defined by any detectable CAD (Gensini > 0) or by moderately obstructive CAD (stenosis in any vessel > 50%) but was significant for severely obstructive CAD defined by stenosis in any vessel > 75% (*p* = 0.017).

### 3.4. sLOX-1 Has Improved Predictive Value in SMuRFless Patients

The percentage of patients who were SMuRFless—i.e., had none of the SMuRFs—was higher in the CAD group (32.8%) when compared to the calcified-plaque predominant group (17.8%) and the soft-plaque predominant group (12.8%). Despite these lower sub-cohort numbers, in linear models the addition of sLOX-1 significantly improved the prediction of all three disease scores, with the β coefficient for the Gensini score increasing from 0.062 to 0.271 (*p* < 0.001), the β coefficient for CACS increasing from 0.047 to 0.173 (*p* = 0.008), and the β coefficient for SPS increasing from 0.072 to 0.271 (*p* < 0.001), as shown in Table 2. Model performance in the SMuRFless population was significantly improved by the addition of sLOX-1, as demonstrated in Table 3, and Figure 4 (blue bars). The improvement in the variance accounted for by the models due to the addition of sLOX-1 was 0.4% to 7.3% for the Gensini score, 0.2% to 3.0% for CACS, and 0.5% to 7.3% for SPS.

Logistic regressions for the SMuRFless population (Table 4) demonstrate that the odds ratio for sLOX-1 was not significant for the prediction of CAD presence as defined by any detectable CAD (Gensini > 0) but was significant for the prediction of moderately obstructive CAD as defined by stenosis in any vessel >50% (*p* = 0.004). The SMuRFless subgroup analysis for severely obstructive CAD was not significant, though the number of patients included in this category was very small (*n* = 8) and the *p* value was borderline (*p* = 0.061).

Bivariate correlations between sLOX-1 and the three disease scores are shown in Figure 6, with significant associations seen in the SMuRFless population for CACS (Pearson Correlation 0.155, *p* = 0.026), Gensini (Pearson Correlation 0.251, *p* < 0.001), and SPS (Pearson Correlation 0.257, *p* < 0.001).

### 3.5. sLOX-1 Is Associated with Age, Cigarette Smoking, and Peripheral Vascular Disease

Serum levels of sLOX-1 did not significantly differ between the sexes or in those with hypertension, hyperlipidemia, diabetes mellitus, a significant family history of ischemic heart disease, previous TIA or stroke, inflammatory arthritis, atrial fibrillation, or in those taking any of the cardiac medications assessed in this study (Table 5). As shown in Figure 7a,b, and Table 5, there was a significantly higher level of sLOX-1 in those who had a smoking history of greater than 10 pack-years (*p* = 0.037) and in those with peripheral arterial disease (*p* = 0.026), despite the small number of patients who had a prior history of peripheral arterial disease (n = 14). In bivariate analysis, serum sLOX-1 level was also weakly associated with increasing age (Pearson correlation 0.065, *p* = 0.029, Figure 7c), but was not associated with body mass index.

## 4. Discussion

In this study, we provide the largest analysis of sLOX-1 as a biomarker for the detection of atherosclerotic CAD that has been performed to date. In this CTCA cohort, we assessed nearly a thousand patients with imaging-quantified CAD, who ranged from having entirely healthy coronary arteries to being severely diseased, across a real-world variety of ages and risk demographics. The ACS cohort was included as a positive control and confirmed the results of previous studies, which showed the sLOX-1 is significantly elevated in ACS [22,23,24].

We have demonstrated in linear regression models that sLOX-1 is associated with disease severity scores that reflect the overall degree of stenosis (Gensini score) and the burden of soft plaque present. sLOX-1 was not associated with increasing coronary calcification alone (CACS). When we looked at logistic regression models relating sLOX-1 to disease incidence, we found no significant relationship with the presence of CAD as defined by any detectable atherosclerosis (Gensini > 0) or by moderately obstructive atherosclerosis with stenosis > 50% in any vessel, but there was a significant association with the presence of severely obstructive CAD (stenosis > 75% in any vessel). This suggests that the elevation of sLOX-1 seen in the disease may relate to the presence of more hemodynamically significant coronary lesions.

These results concur with the limited existing data on sLOX-1 in stable CAD in the literature. One previous study demonstrated a correlation between serum sLOX-1 levels and Gensini score in a cohort of 112 patients with metabolic syndrome who underwent invasive coronary angiography [35], though it is difficult to compare the cohorts as the Gensini scores were not presented with the cohort data, so disease severity is difficult to ascertain and the indication for angiography was unclear. To our knowledge, there are no other studies that relate sLOX-1 levels to angiographic disease scores that have been published to date. Unscored stable CAD was used as a control group in one early study looking at levels of sLOX-1 in a biomarker study for ACS; however, results for the subgroups of interest were below the limit of detection of the custom assay used at that time, so no conclusions can be drawn [23].

The finding that sLOX-1 is more strongly associated with severe disease and a higher burden of soft plaque also fits with the existing data relating sLOX-1 to lesion complexity [27] and instability [25]. There are also numerous studies that demonstrate that sLOX-1 is increased with higher rates of MACE, both following ACS [36,37] and in stable patients [28,38]. These findings could be explained by the association of sLOX-1 with higher-risk soft-plaque lesions, which are more likely to rupture.

To confirm that the association of sLOX-1 with the disease was not secondary to a confounding association with one or more other clinical factors, sLOX-1 levels were examined in patients with different risk factors and who were taking cardiac medications. The only significant associations identified were higher levels of sLOX-1 in patients with a significant smoking history of greater than 10 pack years and in those with a history of peripheral arterial disease. Regression analysis demonstrated that the relationship between sLOX-1 and disease severity persisted despite correction for age and significant smoking history. Peripheral arterial disease was not included in the regression models due to the very low number of patients who had it (n = 14). There was also a weak association between sLOX-1 and increasing age. No significant associations were identified between sLOX-1 and body mass index, sex, hypertension, hyperlipidemia, diabetes mellitus, significant family history of ischemic heart disease, previous TIA or stroke, inflammatory arthritis, atrial fibrillation, or in those taking anti-platelets, anti-coagulants, statins, beta blockers, ACE inhibitors or angiotensin receptor blockers.

The relationships found between sLOX-1 and other cardiovascular risk markers in the literature have been somewhat inconsistent. In smaller studies containing both ACS and non-ACS patients, there has been generally no association found between sLOX-1 levels and age, sex, diabetes, smoking, or hypertension [22,23,28,39,40], and no correlation seen with serum cholesterol (total and LDL) or triglycerides [22,23,28,40]. A weak inverse association was found with HDL in one study [23], but this was not replicated in others. Another study demonstrated higher mean sLOX-1 levels in males and in smokers [24]; however, this was in a patient population that was comprised of greater than 50% ACS patients. This patient demographic likely explains these results, as ACS patients would be over-represented in the male and smoking subgroups, pushing the mean sLOX-1 levels up due to the infarction.

sLOX-1 has previously been associated with peripheral arterial disease in a diabetic population [41], which fits with the results seen in this study. sLOX-1 is also known to be associated with acute stroke [42,43], with significant but asymptomatic carotid atherosclerosis [44], and was predictive of future stroke risk in a large registry study [18]. In our study, we did not identify an increase in the mean levels of sLOX-1 in the patients who had a history of TIA or stroke, though we did not differentiate between embolic and atherosclerotic etiology in our data and there was a relatively low number of patients with that background (n = 50). Overall, our results—in combination with previous studies—suggest that sLOX-1 is a biomarker for atherosclerotic disease of the blood vessels, which is minimally associated with the major cardiovascular risk factors and demographic features that are usually used in primary prevention risk models for CAD.

Finally, in our post hoc sub-analysis of SMuRFless patients, we demonstrated that the addition of sLOX-1 to disease prediction models demonstrated an improved ability to predict disease severity for all three disease scores and to predict the incidence of moderately obstructive CAD. Bivariate analysis also demonstrated a more significant association between sLOX-1 and the raw scores in the SMuRFless population. While the odds ratio for severe obstructive disease in SMuRFless patients was not significant, the analysis at this subgroup level was likely underpowered due to the small number of patients who were both SMuRFless and had severely obstructive CAD in this cohort (n = 8).

The improved prediction of CAD severity and incidence in SMuRFless patients is suggestive that the mechanism that results in the elevation of sLOX-1 may be even more of an important moderator of risk than in those whose disease is driven by traditional risk factors. The importance of the SMuRFless cohort is becoming more apparent as increasing numbers of studies identify these patients across the world [6,7,8,9,10,11] and confirm they have worse outcomes [6,9,10]. It is not currently well understood what drives the generation of atherosclerosis in the absence of traditional risk factors, or whether these patients respond as well to traditional therapy for ischemic heart disease. In this study, we have demonstrated that in SMuRFless patients sLOX-1 is substantially more predictive of disease severity and incidence than either body mass index or a family history of premature coronary artery disease. This fits with previous data, which showed that SMuRFless patients have lower mean BMIs [6,11] and similar rates of family history of premature CAD [7,8]. To our knowledge, this is the first biomarker that has demonstrated a significant association with CAD in the SMuRFless cohort.

This study had several strengths, including the relatively large cohort size and the detailed analysis of the CTCA imaging, which enabled accurate phenotyping of the patient cohort. However, there are some limitations that require consideration. sLOX-1 has been demonstrated to be elevated in atherosclerotic carotid artery disease and lower limb arterial disease, which are not accounted for in our imaging measures. There is a significant correlation between disease burden in the coronaries and other vascular beds [45,46], but further research into sLOX-1 as a CAD biomarker will need to account for atherosclerosis in these other locations. The other main limitation of our study is the binary nature of the risk factor variables, which does not capture the continuous nature of these conditions in terms of severity. However, the consistency of the findings in this study with most of the results from the literature are reassuring that comorbidity severity is unlikely to be driving significant changes in sLOX-1.

In conclusion, we have demonstrated that sLOX-1 is a biomarker for atherosclerotic CAD severity, which demonstrates improved predictive capacity in the SMuRFless population. Mechanistically, this suggests sLOX-1 is important in the poorly understood processes of disease that underpin the development of atherosclerosis in patients with few traditional risk factors, and further study may reveal novel treatment targets related to this biology. As a biomarker, the association between sLOX-1 and early atherosclerosis is not strong enough to be diagnostic in isolation, but these results warrant further research into the incorporation of sLOX-1 into biomarker panels, which many enable the precision diagnosis of atherosclerosis while it is still early enough to implement preventative strategies that can prevent heart attacks.

## Figures and Tables

**Figure 1 biomolecules-13-01187-f001:**
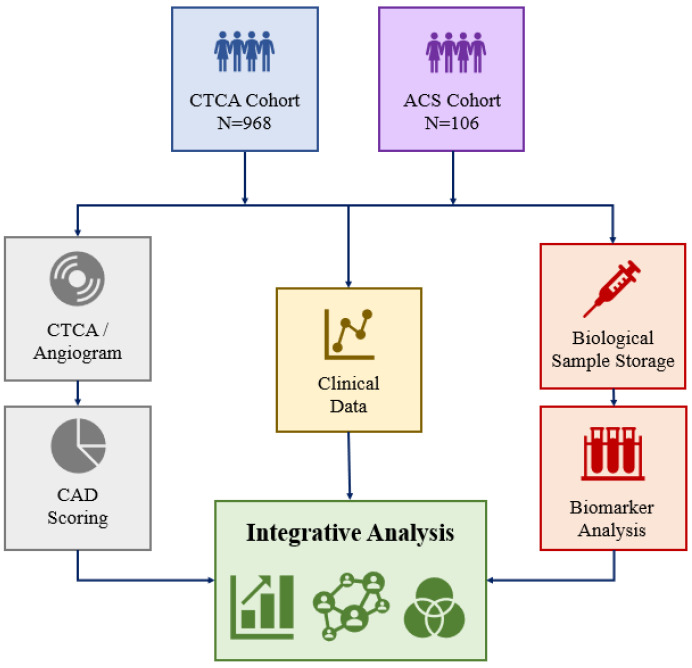
Study design for sLOX-1 biomarker analysis in the BioHEART biobank.

**Figure 2 biomolecules-13-01187-f002:**
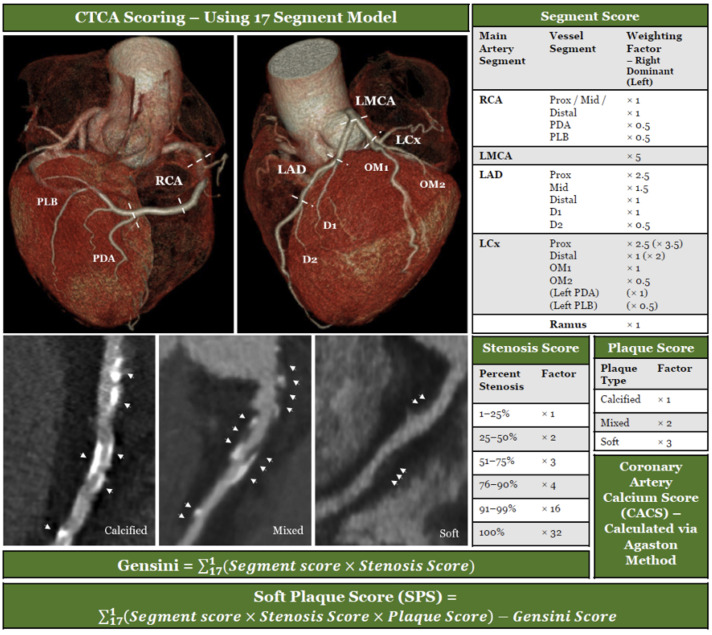
CAD scoring systems used to assess disease burden.

**Figure 3 biomolecules-13-01187-f003:**
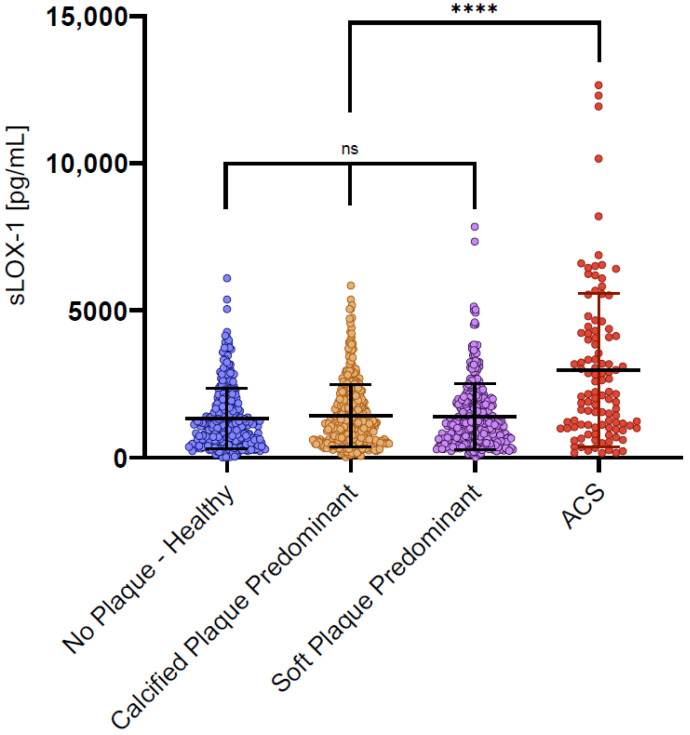
Scatter bar graphs of serum sLOX-1 levels of the 968 patients in the CTCA cohort by disease phenotype subgroup, with comparison to ACS as a positive control. ACS: acute coronary syndrome; sLOX-1: soluble lectin-like oxidized low-density lipoprotein receptor-1. **** *p* < 0.001, ns = not significant.

**Figure 4 biomolecules-13-01187-f004:**
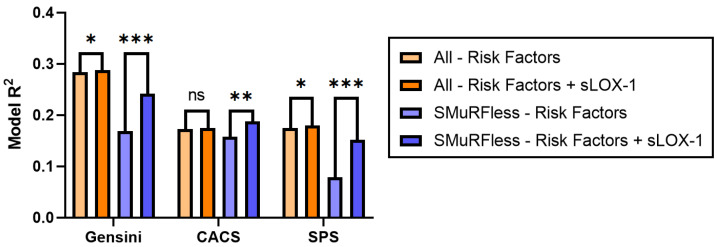
Comparison of model R^2^ values for multi-variable linear regression models, showing differences between models using risk factors (age, sex, body mass index, hypertension, hyperlipidemia, diabetes mellitus, significant smoking history, and significant family history of ischemic heart disease) and models using risk factors plus serum sLOX-1 values; SMuRFless subgroup analysis shown in blue shades. CACS: coronary artery calcium score; SPS: soft-plaque score; sLOX-1: soluble lectin-like oxidized low-density lipoprotein receptor-1. * *p* < 0.05, ** *p* < 0.01, *** *p* < 0.001, ns = not significant.

**Figure 5 biomolecules-13-01187-f005:**
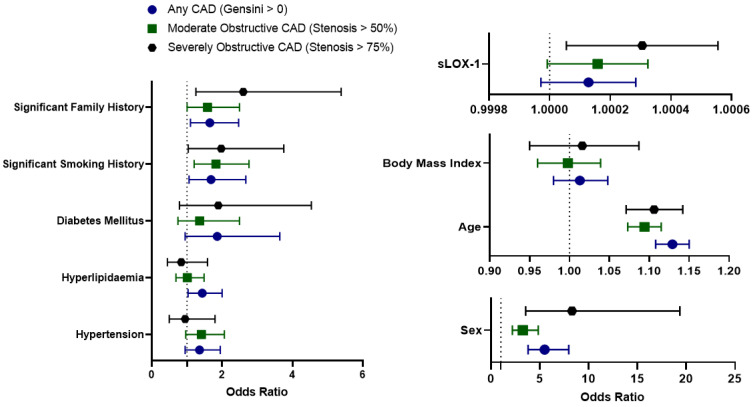
Odds ratios for the incidence of disease for all patients in the CTCA cohort as defined by the presence of any CAD (Gensini score > 0, blue), moderately obstructive CAD (stenosis in any vessel > 50%, green), or severely obstructive CAD (stenosis in any vessel > 75%, black) from multivariable logistic regression models.

**Figure 6 biomolecules-13-01187-f006:**
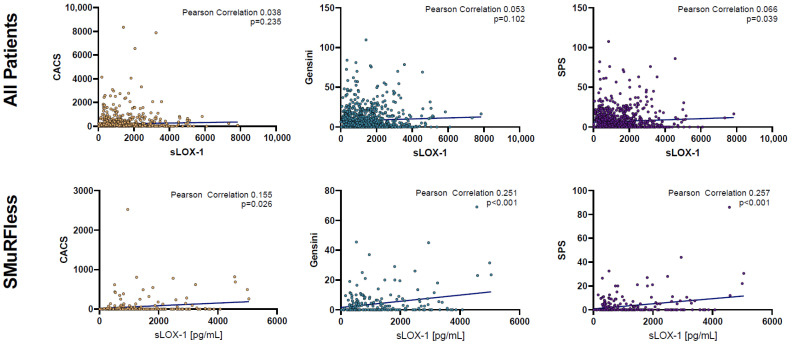
Bivariate correlations between sLOX-1 and the three disease scores: CACS, Gensini, and SPS, for all patients (**top** panel) and SMuRFless patients (**bottom** panel). CACS: coronary artery calcium score; SPS: soft-plaque score; sLOX-1: soluble lectin-like oxidized low-density lipoprotein receptor-1.

**Figure 7 biomolecules-13-01187-f007:**
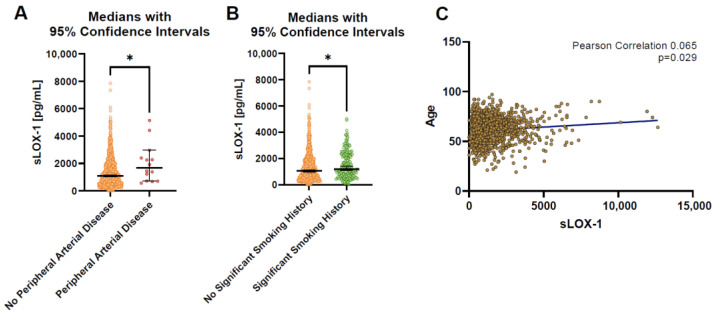
Scatter bar graphs demonstrating significant differences in serum sLOX-1 levels in those with a history of (**A**) peripheral arterial disease or (**B**) a significant smoking history in the CTCA cohort, and (**C**) scatter plot of sLOX-1 vs. age, with trend line demonstrating the bivariate correlation between sLOX-1 and age. * *p* < 0.05.

**Table 1 biomolecules-13-01187-t001:** Cohort Clinical Characteristics, Disease Burden, and Soluble LOX-1 Levels by Subgroup.

Demographics and Risk Factors	All CTCA	No Visible CAD	Calcified Plaque Predominant	Soft Plaque Predominant	Acute Coronary Syndrome
Number—*n*	968	333	309	327	106
Age, years—mean (SD)	61 (12)	53 (11)	63 (11)	67 (10)	64 (13)
Sex, male—*n* (%)	539 (55.7%)	142 (42.8%)	183 (59.2%)	214 (65.4%)	75 (71.4%)
SMuRFs—*n* (%)					
0	206 (21.3%)	109 (32.8%)	55 (17.8%)	42 (12.8%)	20 (19.0%)
1	408 (42.1%)	152 (45.8%)	128 (41.4%)	128 (39.1%)	30 (28.6%)
2	259 (26.8%)	58 (17.5%)	93 (30.1%)	108 (33.0%)	33 (31.4%)
3	82 (8.5%)	13 (3.9%)	29 (9.4%)	40 (12.2%)	20 (19.0%)
4	13 (1.3%)	0 (0.0%)	4 (1.3%)	9 (2.8%)	2 (1.9%)
Hypertension—*n* (%)	379 (39.2%)	91 (27.4%)	134 (43.4%)	154 (47.1%)	48 (45.7%)
Hyperlipidaemia—*n* (%)	572 (59.1%)	158 (47.6%)	187 (60.5%)	227 (69.4%)	59 (56.2%)
Diabetes Mellitus—*n* (%)	85 (8.8%)	21 (6.3%)	25 (8.1%)	39 (11.9%)	18 (17.1%)
Significant Smoking History—*n* (%)	188 (19.4%)	37 (11.1%)	71 (23.0%)	80 (24.5%)	39 (37.1%)
Significant Family History of Premature CAD—*n* (%)	201 (20.8%)	67 (20.2%)	78 (25.2%)	56 (17.1%)	19 (18.1%)
Atrial Fibrillation—*n* (%)	117 (12.1%)	30 (9.0%)	44 (14.2%)	43 (13.1%)	5 (4.8%)
Previous TIA/Stroke—*n* (%)	50 (5.2%)	12 (3.6%)	17 (5.5%)	21 (6.4%)	4 (3.8%)
Peripheral Arterial Disease—*n* (%)	14 (1.4%)	3 (0.9%)	3 (1.0%)	8 (2.4%)	3 (2.9%)
Inflammatory Arthritis—*n* (%)	104 (10.7%)	18 (5.4%)	39 (12.6%)	47 (14.4%)	20 (19.0%)
Anti-Platelet Medication—*n* (%)	171 (17.1%)	44 (13.3%)	52 (16.8%)	75 (22.9%)	19 (18.1%)
Anti-Coagulant Medication—*n* (%)	87 (9.0%)	20 (6.0%)	44 (14.2%)	43 (13.1%)	6 (5.7%)
Statin—*n* (%)	323 (33.4%)	60 (18.1%)	105 (34.0%)	158 (48.3%)	27 (25.7%)
Beta Blocker—*n* (%)	139 (14.4%)	34 (10.2%)	51 (16.5%)	54 (16.5%)	14 (13.3%)
ACE Inhibitor/Angiotensin Receptor Blocker—*n* (%)	309 (31.9%)	66 (19.9%)	106 (34.3%)	137 (41.9%)	28 (26.7%)
Gensini Score—median (IQR)	3.5 (11.5)	0	6.0 (10.0)	11.0 (14.5)	n/a
Soft Plaque Score—median (IQR)	2.5 (9.5)	0	6.8 (15.0)	6.5 (10.0)	n/a
Coronary Artery Calcium Score—median (IQR)	9.9 (146.0)	0	26.7 (121.7)	168.1 (427.8)	n/a
Calcium Percentile, Age & Sex Adjusted—mean (SD)	38.2% (36.2%)	0%	48.8% (31.3%)	67.0% (23.4%)	n/a
Obstructive Disease—*n* (%)					
Moderate (> 50%)	177 (18.3%)	0	63 (20.4%)	114 (34.9%)	109 (100%)
Severe (> 75%)	54 (5.6%)	0	15 (4.9%)	39 (11.9%)	109 (100%)
sLOX-1 [pg/mL]—mean (SD)					
All	1372.9 (1069.2)	1327.2 (1031.2)	1401.0 (1057.6)	1392.8 (1118.7)	3003.1 (2610.6)
SMuRFless	1367.8 (1070.4)	1320.1 (1012.1)	1406.0 (1133.6)	1441.8 (1150.6)	4146.4 (3925.8)

**Table 2 biomolecules-13-01187-t002:** Standardized beta coefficients and *p* values from multi-variable linear regression models showing all patients (top) and SMuRFless patients (bottom) across the three disease scores.

	Gensini		CACS		SPS	
All Patients	Standardized Beta	*p* Value	Standardized Beta	*p* Value	Standardized Beta	*p* Value
Age	0.433	<0.001	0.340	<0.001	0.323	<0.001
Sex	0.307	<0.001	0.238	<0.001	0.251	<0.001
Body Mass Index	−0.019	0.502	−0.022	0.484	−0.021	0.494
Hypertension	0.074	0.012	0.046	0.141	0.073	0.020
Hyperlipidaemia	0.046	0.102	0.045	0.135	0.024	0.434
Diabetes Mellitus	0.059	0.035	0.083	0.006	0.036	0.229
Significant Smoking History	0.062	0.028	0.025	0.415	0.075	0.013
Significant Family History	0.078	0.005	0.044	0.139	0.074	0.013
sLOX-1	0.062	0.025	0.047	0.110	0.072	0.015
**SMuRFless Patients**						
Age	0.367	<0.001	0.335	<0.001	0.258	<0.001
Sex	0.259	<0.001	0.266	<0.001	0.181	0.007
Body Mass Index	0.042	0.500	0.033	0.613	0.009	0.893
Significant Family History	0.005	0.935	-0.039	0.540	0.013	0.845
sLOX-1	0.271	<0.001	0.173	0.008	0.271	<0.001

**Table 3 biomolecules-13-01187-t003:** Comparison of linear regression models predicting disease severity.

	R^2^	Adjusted R^2^	F Value	*p* Value	*p* for Model Change
** *Gensini—All* **					
Risk Factors	0.284	0.278	47.334	<0.001	
Risk Factors + sLOX-1	0.288	0.281	42.815	<0.001	0.025
** *Gensini—SMuRFless* **					
Risk Factors	0.169	0.153	10.192	<0.001	
Risk Factors + sLOX-1	0.242	0.223	12.703	<0.001	<0.001
** *CACS—All* **					
Risk Factors	0.173	0.166	24.984	<0.001	
Risk Factors + sLOX-1	0.175	0.168	22.530	<0.001	0.110
** *CACS—SMuRFless* **					
Risk Factors	0.158	0.141	9.320	<0.001	
Risk Factors + sLOX-1	0.188	0.167	9.144	<0.001	0.008
** *SPS—All* **					
Risk Factors	0.175	0.168	25.338	<0.001	
Risk Factors + sLOX-1	0.180	0.172	23.293	<0.001	0.015
** *SPS—SMuRFless* **					
Risk Factors	0.079	0.061	4.313	0.002	
Risk Factors + sLOX-1	0.152	0.131	7.156	<0.001	<0.001

**Table 4 biomolecules-13-01187-t004:** Odds ratios for the incidence of disease in the CTCA cohort as defined by the presence of any CAD (Gensini score > 0, left column), moderately obstructive CAD (stenosis > 50%, middle column), and severely obstructive CAD (stenosis > 75%, right column) from multivariable logistic regression models.

	Any CAD (Gensini > 0, n = 635)		Moderate Obstructive CAD (Stenosis > 50%, n = 142)		Severe Obstructive CAD (Stenosis > 75%, n = 54)	
All Patients (n = 968)	Odds Ratio (95% CI)	*p* Value	Odds Ratio (95% CI)	*p* Value	Odds Ratio (95% CI)	*p* Value
Age	1.129 (1.108–1.150)	<0.001	1.094 (1.073–1.115)	<0.001	1.106 (1.071–1.142)	<0.001
Sex	5.499 (3.798–7.960)	<0.001	3.255 (2.189–4.841)	<0.001	8.294 (3.555–19.350)	<0.001
Body Mass Index	1.013 (0.980–1.048)	0.445	0.998 (0.960–1.039)	0.940	1.016 (0.950–1.087)	0.638
Hypertension	1.359 (0.949–1.948)	0.094	1.413 (0.966–2.066)	0.075	0.947 (0.499–1.799)	0.868
Hyperlipidaemia	1.436 (1.031–2.002)	0.033	1.013 (0.689–1.490)	0.948	0.840 (0.445–1.586)	0.592
Diabetes Mellitus	1.860 (0.950–3.641)	0.070	1.363 (0.745–2.494)	0.314	1.892 (0.789–4.538)	0.153
Significant Smoking History	1.685 (1.062–2.676)	0.027	1.825 (1.205–2.764)	0.005	1.974 (1.037–3.755)	0.038
Significant Family History	1.651 (1.102–2.473)	0.015	1.584 (1.006–2.495)	0.047	2.601 (1.256–5.386)	0.010
sLOX-1	1.000 (0.999–1.001)	0.110	1.000 (0.999–1.001)	0.062	1.000 (1.000–1.001)	0.017
All Patients: Model R^2^	0.420		0.258		0.269	
**SMuRFless Patients (n = 205)**	**Any CAD** **(Gensini > 0, n = 97)**		**Moderate Obstructive CAD** **(Stenosis > 50%, n = 21)**		**Severe Obstructive CAD** **(Stenosis > 75%, n = 8)**	
Age	1.109 (1.073–1.145)	<0.001	1.092 (1.046–1.141)	<0.001	1.057 (0.995–1.123)	0.073
Sex	4.251 (2.048–8.825)	<0.001	4.263 (1.280–14.203)	0.018	6.111 (0.699–53.452)	0.102
Body Mass Index	0.997 (0.933–1.065)	0.936	1.064 (0.948–1.194)	0.295	1.096 (0.926–1.296)	0.286
Significant Family History	1.533 (0.712–3.303)	0.275	0.674 (0.189–2.409)	0.544	4.120 (0.885–19.192)	0.071
sLOX-1	1.000 (0.999–1.001)	0.139	1.001 (1.000–1.001)	0.004	1.001 (0.999–1.001)	0.061
SMuRFless: Model R^2^	0.364		0.317		0.243	

**Table 5 biomolecules-13-01187-t005:** Comparison of sLOX-1 levels in patients from the CTCA cohort showing differences between those with and without risk factors, and who were or were not taking cardiac medications.

	N	Mean	SD	*p* Value
Male	539	1320.75	1043.57	0.089
Female	429	1438.37	1098.38	
Hypertension (−)	589	1364.32	1038.37	0.756
Hypertension (+)	379	1386.18	1116.80	
Hyperlipidaemia (−)	396	1338.73	1039.49	0.409
Hyperlipidaemia (+)	572	1396.52	1089.65	
Diabetes Mellitus (−)	833	1371.82	1046.09	0.921
Diabetes Mellitus (+)	85	1383.83	1292.73	
Significant Smoking History (−)	780	1337.77	1077.30	0.037
Significant Smoking History (+)	188	1518.55	1025.12	
Significant Family History (−)	767	1359.94	1071.22	0.462
Significant Family History (+)	201	1422.26	1062.87	
Transient Ischaemic Attack or Stroke (−)	918	1369.15	1067.60	0.642
Transient Ischaemic Attack or Stroke (+)	50	1441.36	1107.83	
Peripheral Arterial Disease (−)	954	1363.62	1061.88	0.026
Peripheral Arterial Disease (+)	14	2003.85	1395.18	
Inflammatory Arthritis (−)	864	1384.77	1091.01	0.319
Inflammatory Arthritis (+)	104	1274.08	865.68	
Atrial Fibrillation (−)	851	1383.22	1079.63	0.417
Atrial Fibrillation (+)	117	1297.63	991.54	
Statin (−)	645	1356.00	1067.90	0.489
Statin (+)	323	1406.59	1072.81	
Anti-Coagulant (−)	881	1365.84	1061.64	0.515
Anti-Coagulant (+)	87	1444.19	1147.48	
Anti-Platelet (−)	797	1363.56	1075.33	0.559
Anti-Platelet (+)	171	1416.30	1042.41	
Beta Blocker (−)	829	1378.92	1073.52	0.668
Beta Blocker (+)	139	1336.87	1046.49	
ACE Inhibitor or Angiotensin Receptor Blocker (−)	659	1357.40	1018.41	0.514
ACE Inhibitor or Angiotensin Receptor Blocker (−)	309	1405.67	1171.56	

## Data Availability

The data that support this study are available from the corresponding author upon reasonable request.

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
