# Peer review of "Serum Soluble Lectin-like Oxidized Low-Density Lipoprotein Receptor-1 (sLOX-1) Is Associated with Atherosclerosis Severity in Coronary Artery Disease"

_biomolecules, 2023, doi:10.3390/biom13081187_

Round 1

Reviewer 1 Report

The idea of associating the increase of some biomarkers with the destabilization of coronary atheroma plaques or with the earliest possible diagnosis of ischemic heart disease is up to date and an interesting one.

The association of the destabilization of atheroma plaques with the increase of inflammatory biomarkers (myeloperoxidase, interleukins, etc.) is already documented in the literature.

Current research brings to the foreground a much more interesting association between sLOX-1 and coronary heart disease.

An impressive number of patients were involved in this research - 968 participants according to the protocol for the BIOHEART biobank from Australia and New Zealand.

The methodology of the study is extremely meticulous and well argued.

The results of their study are remarkable, the main conclusion according to which sLOX-1 is associated with CAD severity and with risk prediction in the SMuRFless population is of great scientific value, with possible implications in current clinical practice.

Of course, as the authors also showed, new studies are needed to incorporate sLOX-1 into biomarker panels, useful in the early diagnosis of CAD and the prevention of heart attacks.

Author Response

Many thanks to the reviewer for their kind comments. As no changes have been requested, we have not made any alterations to the manuscript in response.

Reviewer 2 Report

In this manuscript (Ms ID: biomolecules-2454254), the authors hypothesized that serum sLOX-1 would be independently associated with coronary artery disease (CAD). To test this hypothesis, the authors quantified serum sLOX-1 via ELISA from 968 participants with CT coronary angiograms in the BioHEART study and linked serum sLOX-1 with disease incidence and severity which was scored in the form of coronary artery calcium scores (CACS), Gensini scores, and a semi-quantitative soft-plaque score (SPS). Finally, the authors concluded that sLOX-1 is associated with CAD severity and can be used as a biomarker for risk prediction in the population of the standard modifiable cardiovascular risk factors (SMuRFs). This study is simple but the results may be useful for the prediction of CAD or for developing a novel biomarker. The manuscript is well organized, and I have the following minor comments.

1.     In the manuscript Title, “EARLY coronary artery disease” is mentioned. However, the Abstract does not mention “EARLY”. Please either delete the word “EARLY” in the Title or mention it in the Abstract.

2.     In Keywords (line 33), full names should be provided for the abbreviations, i.e., CTCA, CAD, sLOX-1.

3.     Lines 218-219: “Consistent with prior studies, …” Please provide references for the prior studies.

4.     Line 313: “soft plaque present (SPS)”. The phrase seems not correspond to SPS.

Author Response

We appreciate the positive comments made by the reviewer and have addressed the reviewer’s minor comments point-by-point below.  

  1. In the manuscript Title, “EARLY coronary artery disease” is mentioned. However, the Abstract does not mention “EARLY”. Please either delete the word “EARLY” in the Title or mention it in the Abstract.
    • Thank you for pointing out this discrepancy – we agree that the word "early" should be removed from the title.
  2. In Keywords (line 33), full names should be provided for the abbreviations, i.e., CTCA, CAD, sLOX-1.
    • We have added the full terms for each abbreviation in the keyword list.
  3. Lines 218-219: “Consistent with prior studies, …” Please provide references for the prior studies.
    • Thank you for pointing out this omission, the relevant references have now been included on this line.
  4. Line 313: “soft plaque present (SPS)”. The phrase seems not correspond to SPS.
    • Many thanks for identifying this error, the reference to SPS has been removed.